# Robust Pulse Rate Measurements from Facial Videos in Diverse Environments

**DOI:** 10.3390/s22239373

**Published:** 2022-12-01

**Authors:** Jinsoo Park, Kwangseok Hong

**Affiliations:** 1Department of Electrical and Computer Engineering, Sungkyunkwan University, 2066 Seobu-ro, Jangan-gu, Suwon-si 16419, Republic of Korea; 2School of Electronic Electrical Engineering, Sungkyunkwan University, 2066 Seobu-ro, Jangan-gu, Suwon-si 16419, Republic of Korea

**Keywords:** pulse rate, pulse wave, non-contact, facial video, stable face detection, color difference component, diverse environment

## Abstract

Pulse wave and pulse rate are important indicators of cardiovascular health. Technologies that can check the pulse by contacting the skin with optical sensors built into smart devices have been developed. However, this may cause inconvenience, such as foreign body sensation. Accordingly, studies have been conducted on non-contact pulse rate measurements using facial videos focused on the indoors. Moreover, since the majority of studies are conducted indoors, the error in the pulse rate measurement in outdoor environments, such as an outdoor bench, car and drone, is high. In this paper, to deal with this issue, we focus on developing a robust pulse measurement method based on facial videos taken in diverse environments. The proposed method stably detects faces by removing high-frequency components of face coordinate signals derived from fine body tremors and illumination conditions. It optimizes for extracting skin color changes by reducing illumination-caused noise using the Cg color difference component. The robust pulse wave is extracted from the Cg signal using FFT–iFFT with zero-padding. It can eliminate signal-filtering distortion effectively. We demonstrate that the proposed method relieves pulse rate measurement problems, producing 3.36, 5.81, and 6.09 bpm RMSE for an outdoor bench, driving car, and flying drone, respectively.

## 1. Introduction

Pulse rate is an essential indicator of cardiovascular health. It can predict the incidence, progression, and mortality associated with cardiovascular disease (CVD). It can also contribute to the daily prevention of CVD and the evaluation of the rehabilitation progress of chronic diseases [1,2,3,4]. In hospitals and clinical institutions, physiological signals are traditionally used to provide users with valuable information about the cardiovascular system, including pulse, blood pressure, arterial oxygen saturation, breathing, and cardiac autonomic functions [5,6].

The importance of preventing diseases in daily life and the pursuit of human health are constantly increasing. In line with the increased interest in personal health, wearable and non-wearable technologies are also developing worldwide [4,7,8]. However, wearable technology may cause problems in pulse rate monitoring due to incomplete contact between the sensor and the skin [2,9]. The user must be in long-term contact with the sensor for monitoring, and it is necessary to continuously check whether the sensor is operating in tight contact with the user. The number of sensors must also match the number of individuals who are required to be measured. Therefore, wearable technology is unsuitable for certain individuals and environments, including newborns, drivers, exercisers, skin-damaged patients and disaster-affected patients [10].

To solve this problem, researchers are examining non-contact methods for measuring the pulse using facial video, which minimizes user intervention and enables monitoring of the pulse rate when driving [11], sleeping [12], or stress monitoring [13] is required.

The development of a non-contact method for measuring pulse rate that is quick, simple, and feasible in real-time has resulted in the creation of new applications for use in various fields [14,15]. The primary objective of non-contact pulse measurement studies is to analyze skin color changes in relation to heartbeats and facial movements using built-in cameras on smart devices such as mobile phones and tablets.

Previous research has demonstrated various methods for measuring pulse rate by applying independent component analysis (ICA), principal component analysis (PCA), fast Fourier transform (FFT), band pass filter (BPF) to RGB color data calculated in the region of interest (ROI) of facial images [1,7,16,17,18,19,20,21,22,23,24,25,26,27,28,29,30,31,32,33,34], and by analyzing head movement and blinking [12,14,35] under controlled laboratory conditions. However, in external environments (outdoor bench, car, drone), the pulse rate cannot be accurately measured due to factors such as the user’s fine body tremor, illumination changes, non-detection of the face, etc.

In this paper, methods are presented for accurately measuring the robust pulse rate in diverse environments. In summary, our procedures are as follows.

We propose a method for stable face detection by removing high-frequency components of face coordinate signals derived from fine body tremors and illumination conditions. A moving average filter (MAF) is applied to a specified number of X and Y coordinate samples for the detected face coordinates when a face is detected in facial videos collected in diverse environments. Then, the skin ROI is identified based on the facial coordinates that have been detected reliably. Through this method, it is possible to effectively solve the problem of uncalculated skin color values resulting from non-detection of the face in an external environment.

Further, we use the Cg color difference component to optimize the extraction of skin color changes by minimizing illumination-induced noise. The RGB color system of the ROI is converted into the YCgCo color system in order to extract the Cg signal over time. The purpose of using Cg color is that it is optimized for detecting changes in skin color. R, G and B colors are unsuitable for extracting the pulse wave in diverse environments due to the fact that their values vary significantly under various illumination conditions, such as incandescent lighting, fluorescent lighting and sunlight [36,37].

We propose a method for the robust pulse wave extraction from the Cg signal calculated by the ROI of skin in diverse environments. Pulse rate measurement relies on the pulse wave, which can be extracted from contact-type PPG and ECG devices. To extract the pulse wave using the Cg signal, FFT is applied to the Cg signal calculated from the facial video to convert it into a frequency domain, and zero-padding is applied to all bands with the exception of those related to the pulse (0.67~3.34 Hz). Then, it is extracted by applying inverse FFT (iFFT) to the frequency domain computed by the zero-padding. Signal-filtering distortion is effectively eliminated with this method.

The proposed method for measuring the robust pulse rate in diverse environments uses the average of the RR interval (interval between peak points) calculated from the pulse wave signal.

## 2. Related Work

Remote non-contact pulse rate measurements are appealing for both commercial and academic applications [21,26]. Several papers from the past few years have proposed a color-based method for non-contact pulse rate measurements using general cameras to solve the problem of contact-type measuring devices [12,14,38]. Existing studies have commonly specified facial region detection and ROI. The pulse wave is subsequently calculated using RGB color channels extracted from facial videos taken by cameras. There are numerous ways to calculate pulse wave and rate. The methods of calculating pulse rate can be summarized as follows: (1) by applying ICA to RGB color data calculated in ROI and FFT to pulse-related frequency bands [1], (2) by detecting a peak point on a signal calculated by applying ICA [33], (3) by detecting a peak point on a pulse wave calculated by applying BPF by setting a cut-off frequency in a pulse-related frequency band [10], (4) by applying PCA to RGB color data calculated in ROI and FFT to pulse-related frequency bands [26], etc. When measuring pulse rates using color data of ROI from facial images, the majority of previous studies have allowed subjects to remain stationary and have only considered a constant indoor environment with minimal illumination changes. As a result of a controlled environment producing simple and nearly noise-free data, all reported results have achieved high precision. However, if the pulse rate is measured in external environments (cafes, mountains, etc.), cars, and drones, cameras can continue to shake when detecting the user’s face, and illumination can also change dramatically. Very few studies have determined how these methods are performed on challenging data when both illumination changes and the subject’s fine body tremor are included. To consider the problem of illumination changes in pulse rate measurements using facial videos, previous studies used methods of: (1) taking a facial video in a limited situation where light sources (LED, natural light, etc.) and subjects were immobilized [1,5,18,20,26], (2) using H, S, and V color data calculated by converting RGB color models of detected ROI to HSV color models, (3) using each component of the separated RGB by applying ICA to the RGB color model [12,26,33]. In order to solve the problem associated with the subject’s fine tremor, a specific number of landmarks were set in the detected face area, and a pulse rate measurement experiment was conducted using RGB color data from the ROI detected in the set face [10,18,19,21]. In addition, none of the data is accessible to the public, and new methods must provide new datasets; thus, repetitive data collection wastes time. Moreover, differences between databases make it difficult to compare methods objectively.

## 3. Methods

### 3.1. Process of Robust Pulse Wave Extraction

This paper aims to measure pulse waves and pulse rates using facial videos that are robust in diverse environments. The robust pulse rate measurement method proposed in this study consists of the following three-steps: pre-processing, extraction of the robust pulse wave, and pulse rate measurement. In the first step, we stably detected faces from a facial video. We also specified the ROI (region of interest) in the face area detected stably in the facial image and calculated the Cg color signal by converting the RGB color scheme of ROI into the YCgCo color scheme. In the next step, the robust pulse wave was calculated by applying FFT and inverse FFT (iFFT) to the calculated Cg signal in pulse-related frequency bands. In the last step, we proposed a method to calculate robust pulse rates in diverse environments using the average of interval values between detected peak points (RR interval) from calculated pulse waves. Figure 1 illustrates the flowchart of the proposed method.

#### 3.1.1. Stable Face Detection

The most important aspect of research on pulse rate measurement using facial video was the development of a technology that can measure pulse rate anytime, anywhere using popular smart devices, for people who have difficulty gaining access to hospitals or medical facilities. You only look once (YOLO), Haar-like feature, histogram of oriented gradient (HOG), single-shot detection (SSD), media pipe, etc., are representative face detection algorithms. In this paper, while conducting a smartphone-based pulse rate measurement, we applied a face detection method using Haar, SSD, and YOLO. The most important aspect of the pulse rate measurement using a smart device-based facial video is the face detection rate, which is proportional to detecting more skin color changes per second. Experiments using a smartphone (Galaxy S10) with a 30 fps front camera revealed an average of 28 fps for Haar, 22 fps (73%) for SSD, and 17 fps (56%) for YOLO. We conducted a facial video-based pulse rate measurement using the Haar feature on the smart device with the highest face detection rate per second.

As shown in Figure 2, the Haar feature is a method for detecting objects based on brightness differences in regions of interest [39]. It performs differently depending on the characteristics of the objects. However, it demonstrates a rapid computational speed and a high accuracy with the face. The face is detected using the brightness difference between the eyes (Figure 2d) and the nose area and the brightness difference between the brows (Figure 2e).

X and Y coordinates of the detected face using the Haar feature were entered at 30 frames per second and the coordinate position was changed from frame to frame. X coordinates were unstably detected over time, as shown in Figure 3. This instability affected the cheek area’s coordinates based on plane coordinates, which hindered the extraction of color values from the identical area of each frame.

To establish a stable ROI, the high-frequency component of the detected plane’s coordinates must be eliminated in real-time. A moving average filter (MAF) was utilized to stabilize each frame’s rapidly changing coordinate value.

As shown in Figure 4, n samples of X or Y coordinates detected with facial images can be applied to formula (1) to compensate for the problem that detected face coordinates are calculated somewhat irregularly from frame to frame without moving the face. In formula (1), ${\bar{x}}_{k}$ represents the X (or Y) coordinates of the corrected face, and $x_{k}$ represents the X (or Y) coordinates of the actual detected face.
(1)$$\bar{x_{k}\;}=\frac{x_{k-n+1}+x_{k-n+2}+x_{k-n+3}+\ldots +x_{k}}{n}$$

The corrected x-coordinate conversion before an application is shown in Figure 5. Compared with Figure 3, the coordinate value showed a gentle change after application. This can extract stable pulse waves with the proposed method and improve pulse rate measurement accuracy when measuring pulse rate outdoors using a smartphone or tablet by hand, or even in external environments such as when a driving car or flying a drone.

#### 3.1.2. Setting the Skin ROI

In this paper, the first step in extracting a pulse wave from a skin region was to define the region of interest (ROI) where the user’s face could be detected from the camera input of a smart device and the pulse wave could be extracted most approximately to the PPG. The majority of smart devices are capable of acquiring 30 images per second, which does not interfere with the pulse wave extraction process proposed in this paper. The typical human pulse rate ranges from 40 to 200 times. The range of frequencies is up to 3.34 Hz. Consequently, a camera on a standard smart device with a sampling rate of 30 Hz can detect a pulse state within the pulse wave.

The pulse wave for calculating the pulse rate was extracted from the color value of the cheek area of the face. As shown in Figure 6, the cheek region has many arteries and a facial artery with the largest diameter of the facial blood vessels. Accordingly, a pulse wave could be obtained using the calculated color data after setting a cheek area as an area of interest in a facial image absorbing natural light or illumination.

Since illumination changes can affect the pulse rate calculation accuracy [15,40], the areas of the ROI are designated on both cheeks ($ROI_{1}$, $ROI_{2}$), as shown in Figure 7. The amount of illumination changes applied to both cheeks can be considered according to environmental changes. A high accuracy pulse rate may be calculated by applying a weighted average to the pulse rate calculated in both cheeks,
(2)$$x_{ROI_{1}}=X+\left(W\times 0.3\right),\;x_{ROI_{2}}=X+\left(Y\times 0.7\right)\;y_{ROI_{1}\left(or\;ROI_{2}\right)}=Y+\left(H\times 0.5\right)\;w_{ROI_{1}\left(or\;ROI_{2}\right)}=\left(W\times 0.08\right),\;h_{ROI_{1}\left(or\;ROI_{2}\right)}=\left(W\times 0.08\right)\;$$
and setting the region of interest in the cheek in the facial coordinates that are set with time. In the cheek area, $x_{ROI_{1}}$ and $x_{ROI_{2}}$ of the x-coordinate represent 30% and 70% of the width from the facial coordinate origin, and the y-coordinate is set at 50% of the facial height. The width and height of the cheek area are set to 8% of the face width, as given by formula (2).

#### 3.1.3. Extraction of Robust Pulse Wave

The average Cg color of the ROI of the face is used to extract the pulse wave from the facial skin image. The Cg color is verified to be superior to R, G and B. Using only the R, G, and B row data of the ROI to calculate the pulse rate using facial images can reduce the pulse measurement accuracy because it is more affected by the illumination change than the detection of the skin color change, which is the most important factor for pulse rate calculation. Mathematically, the temporal $Cg^{i}$ signal corresponding to ith of the ROI is given by formula (3):(3)$$Cg^{i}\;signal=\left[cg_{1}^{i},cg_{2}^{i},\cdots ,\;cg_{n}^{i}\right]$$
where f is the total number of frames and the $cg_{k}^{i}$ color conversion in R, G and B colors is given by formula (4):(4)$$cg_{k}^{i}=-0.25\times {\bar{r}}_{k}^{i}+0.5\times {\bar{g}}_{k}^{i}-0.25\times {\bar{b}}_{k}^{i}$$
where ${\bar{r}}_{k}^{i}$, ${\bar{g}}_{k}^{i}$ and ${\bar{b}}_{k}^{i}$ represent the overall average of R, G and B color values calculated in each pixel in the kth frame for the ith ROI, which is given by formula (5):(5)$${\bar{r}}_{k}^{i}=\frac{1}{p^{i}}\underset{\left(x,\;y\right)\in R^{i}}{\sum }r_{k}\left(x,\;y\right)$$
where $p^{i}$ is the total number of pixels in the ith ROI, $R^{i}$ is the ith ROI, $\left(x,\;y\right)$ is a pixel location and $r_{k}$ represents the R color value in kth frame. ${\bar{g}}_{k}^{i}$ and ${\bar{b}}_{k}^{i}$ are calculated to be equal to ${\bar{r}}_{k}^{i}$.

Before calculating the pulse wave; in the Cg signal shown in Figure 7, pulse-induced pulse waves are detected. In the frequency domain shown in Figure 8, a pulse frequency of 1.256 Hz was confirmed in the pulse-related frequency band of 0.67~3.34 Hz.

After applying iFFT to the data in a frequency domain limited to the pulse-related frequency band of Figure 8, it was possible to calculate pulse waves that were strong in various environments by converting them into a time domain. This does not require a consideration of delay or distortion caused by the filter of pulse waves calculated by applying BPF to R (or G or B) color data in existing studies.

Figure 9 shows the process of pulse wave extraction by applying FFT–iFFT based on zero-padding. The robust pulse wave is calculated by applying iFFT to the frequency domain, in which zero-padding is applied. Zero-padding is given by the following formula (6):(6)$$x_{zp}\left(n\right)=x\left(n\right),\;f_{pulse\_min}\leq n\leq f_{pulse\_max}\;=0,\;\;\;\;\;n<f_{pulse\_min},\;n>f_{pulse\_max}$$
where x represents frequency values calculated by applying FFT to the Cg signal, n is the total number of samples and $x_{zp}$ denotes frequency values to which zero-padding is applied. Zero-padding is performed for the entire area except for the pulse-related frequency band ($f_{pulse\_min}$~$f_{pulse\_max}$ (Hz)). iFFT when applying zero-padding to the frequency values is given by formula (7):(7)$$X\left(n\right)=\frac{1}{N}\overset{N-1}{\underset{k=0}{\sum }}x_{zp}\left(k\right)\times e^{j\left(\frac{2\pi }{N}\right)nk}\;,\;\left(n=0,1,\cdots ,N-1\right)$$

### 3.2. Robust Pulse Rate Measurement Method

We propose a method to measure pulse rates using the RR interval of pulse waves calculated through Figure 9 to measure pulse rates with a high accuracy in facial images. This section describes peak point detection, RR interval calculation, and pulse rate calculation using the RR interval for the proposed method, and further presents four methods (STFT), auto correlation function (ACF), and combined ICA and BPF to evaluate the proposed method.

#### 3.2.1. Peak Detection of Pulse Wave

To calculate the RR interval from the pulse wave, the peak point at the top of the pulse wave must be detected, as shown in Figure 10. In existing studies [9,27,30], the pulse rate is estimated using the peak number of the pulse wave, which is an important characteristic for calculating the pulse rate.

#### 3.2.2. Pulse Rate Measurement Using the RR Interval

The RR interval is used not only for heart rate variation (HRV) analysis, but also for pulse-related condition analysis, such as arrhythmia. The RR interval is the time difference between the previous peak and the next peak point of the pulse wave. In the case of simply calculating a pulse rate using the frequency of a pulse wave, unstable detection (or non-detection) of a peak point may be greatly affected by the pulse rate, so this paper calculates the pulse rate using the average of the RR interval. Figure 11 shows the calculation process of the RR interval.
(8)$$rri_{i}=t\left(peak_{i}\right)-t\left(peak_{i-1}\right),\;i=1,2,\cdots ,n$$
where $rri_{i}$ represents the ith of the RR interval, and the unit is seconds; $peak_{i}$ is the ith of the peak; t is the time function; $t\left(peak_{i}\right)$ represents the time of the ith peak, and the unit is seconds. The average of $rri_{i}$ ($\bar{RRI}$) of the calculated pulse wave is shown in the following formula (9):(9)$$\bar{RRI}=\frac{1}{n}\overset{n}{\underset{i=1}{\sum }}rri_{i}$$

The RFPR (robust facial pulse rate) proposed in this paper uses the $\bar{RRI}$ calculated by $rri_{i}$, which is given by the formula (10):(10)$$\mathrm{RFPF}\left(per\;Minute\right)=\frac{60}{Average\;of\;RR-interval}$$

### 3.3. Pulse Rate Measurement Method Using STFT

As another method of pulse rate estimation, the pulse rate estimation was performed by applying STFT (short-time Fourier transform) to the Cg signal calculated in the ROI, except for the proposed method. The Cg signal is subdivided into regular intervals, as shown in Figure 12.

STFT was applied to the subdivided Cg signal to calculate the pulse from the overall average of frequency values with the greatest power in the frequency domain within the pulse-related frequency band. The average of the largest frequency power for each section is given by the following formula (11):(11)$$\bar{f}=\frac{1}{n}\overset{n}{\underset{i=1}{\sum }}f_{max}^{i}$$
where $\bar{f}$ represents the average of the largest frequency power for each section, $f_{max}^{i}$ is the largest of the frequency power at ith of the section, and n is the total number of the section. We measure the pulse rate using $\bar{f}\;$ calculated by $f_{max}^{i}$, which is given by the formula (12):(12)$$FPR\left(per\;Minute\right)\;using\;STFT=\bar{f}\times 60$$

### 3.4. Pulse Rate Measurement Method Using ACF

As another method of pulse rate estimation, the pulse rate estimation was performed by applying the ACF (auto correlation function) to the pulse wave calculated in the ROI, except for the proposed method. The pulse wave is subdivided at regular intervals to apply the ACF, and the pulse rate is calculated using average intervals of the detected peak at the signal, as shown in Figure 13.
(13)$$ppi_{i}=peak_{i}-peak_{i-1},\;i=1,2,\cdots ,n$$
where $ppi_{i}$ represents the ith of the peak interval, and the unit is the sample rate. $peak_{i}$ is the ith of peak. The average of $ppi_{i}$ ($\bar{PPI}$) of the signal calculated by applying the ACF to the pulse wave is shown in formula (14):(14)$$\bar{PPI}=\frac{1}{n}\overset{n}{\underset{i=1}{\sum }}ppi_{i}$$

We measure the pulse rate using the $\bar{PPI}$ calculated by $ppi_{i}$ and the Fps of the facial video, which is given by (15):(15)$$\mathrm{FPF}\left(per\;Minute\right)\;using\;ACF=\frac{60\times Fps\;of\;Facial\;Video}{Average\;of\;Peak\;interval}$$

### 3.5. Pulse Rate Measurement Method Using ICA and BPF for RGB Color Data

Pulse rate estimation is also performed by applying BPF and ICA to R, G, and B color data calculated from the ROI of the facial video. RGB components are derived by applying ICA to R, G and B color signals calculated in the ROI. This is shown in Figure 14.

BPF is applied to the calculated components to calculate the pulse wave of each of R, G, and B, and the pulse rate is estimated by detecting the peak point from each calculated pulse wave. Figure 15 shows R, G, and B pulse waves calculated by applying BPF to R, G, and B components, compared with time-synchronized PPG signals, and we use G pulse waves to evaluate the performance of the method proposed in this paper.

### 3.6. Weighted Average Pulse Rate for ith of ROI

In this paper, a weighted average was applied to pulse rates calculated from the ROI of both cheeks ($ROI_{1},ROI_{2}$) to calculate pulse rates that are robust to illumination changes. The amount of change in illumination applied throughout the skin according to environmental changes may be considered. The $Facial\;Pulse\;Rate_{wa}$ calculated by applying the weighted average is given by formula (16):(16)$$FPR_{wa}=w_{1}\times FPR_{roi_{1}}+w_{2}\times FPR_{roi_{2}}+\cdots +w_{n}\times FPR_{roi_{n}}$$

As shown in Figure 16, the ROI in the facial image may be from various regions other than both cheeks. In the experimental results, the $w_{1}$ and $w_{2}$ were set to 0.5 and the experiment was conducted.

## 4. Results

### 4.1. Recorded Dataset

In this paper, the proposed pulse measurement method evaluated its performance against the five methods for facial video-based pulse measurements with PPG devices commonly used in medical facilities, such as hospitals and nursing homes.

#### 4.1.1. Indoor Dataset

In the case of an indoor environment, a total of 375 facial videos for a video length of 60 s were collected from 25 different subjects (20 males and 5 females). As shown in Figure 17, videos were acquired from smartphone cameras (Galaxy S10+). Subjects took facial videos while wearing a PPG device (PPG signal calculation and facial video acquisition were time-synchronized and simultaneously performed). The videos obtained thus had a resolution of 640 × 480 pixels and a frame rate of 30 fps.

#### 4.1.2. External Dataset

In an outdoor environment, a total of 60 face videos with a video length of 60 s were collected from six different subjects (five males and one female) while driving cars and flying drones to conduct pulse rate measurement experiments using facial videos. As shown in Figure 18 and Figure 19, subjects took facial videos using the camera from driving cars and flying drones while wearing PPG devices. The signal measured in the PPG device was collected using Bluetooth transmission.

In this paper, as shown in Figure 20, a user’s face color data were extracted using the camera of a smartphone attached to the drone to measure the pulse rate. However, there are cases where it was impossible to detect the face due to the influence of the drone shaking and the surrounding lighting environment. The face color might not be completely extracted because the face is not detected, as shown in Figure 21. Thus, interpolation is needed.

When a face region is detected when flying a drone or driving a car, the skin region of interest is set, a Cg signal is calculated, and an undetected region is generated. To solve this problem, a cubic spline interpolation method was used in this paper. The process is shown in Figure 20. The cubic spline interpolation is given by formulas (17) and (18).

A cubic spline is an algorithm that smoothly connects a given point. This is because the curve connecting the two points is a cubic polynomial ($a_{0}+a_{1}x+a_{2}x^{2}+a_{3}x^{3}$). In other words, we need to connect two points that are apart.

We have five blue points. Cubic Spline interpolation can be used to obtain the data value of $x_{i+1}$ in $x_{i}$ and to connect smooth curves. That is, the function values of the two curves at each point must be the same, and the derivative of the two curves at each point $x_{n}$ must also be the same.
(17)$$\;S_{i}=f\left(x_{i}\right)\;and\;S_{i}\left(x_{i+1}\right)=f\left(x_{i+1}\right)\;for\;each\;i=0,1,\cdots ,n-2\;\;S_{i+1}\left(x_{i+1}\right)=S_{i}\left(x_{i+1}\right)\;\;for\;each\;i=0,1,\cdots ,n-2\;S_{i+1}^{\prime }\left(x_{i+1}\right)=S_{i}^{\prime }\left(x_{i+1}\right)\;\;for\;each\;i=0,1,\cdots ,n-2\;S_{i+1}^{\prime \prime }\left(x_{i+1}\right)=S_{i}^{\prime \prime }\left(x_{i+1}\right)\;for\;each\;i=0,1,\cdots ,n-2$$
(18)$$S^{\prime \prime }\left(x_{0}\right)=S^{\prime \prime }\left(x_{n}\right)=0\;\;S^{\prime \prime }\left(x_{0}\right)=f^{\prime }\left(x_{0}\right)\;and\;S^{\prime \prime }\left(x_{n}\right)=f^{\prime }\left(x_{n}\right)$$

The $S_{i}$ is converted into a third-order polynomial by applying a Cg signal with undetected regions to the proposed formulas (17) and (18), the undetected part can be solved as shown in the interposed Cg signal in Figure 20, and the pulse rate can be calculated.

### 4.2. Performance Evaluation

The performance evaluation used in our experiments is based on the calculated FPR error ($y_{i}-{\hat{y}}_{i}$), where $y_{i}$ denotes the actual pulse rate calculated from ith of the PPG and ${\hat{y}}_{i}$ represents the facial pulse rate (FPR) calculated from *i*th of the facial video. The evaluation includes MAE (mean absolute error), RMSE (root mean square error), and MAPE (mean absolute percentage error), which are given by formula (19):(19)$$MAE=\frac{1}{n}\sum_{i=1}^{n}\left|y_{i}-{\hat{y}}_{i}\right|,\;RMSE=\sqrt{\sum_{i=1}^{n}\frac{{\left(y_{i}-{\hat{y}}_{i}\right)}^{2}}{n}},\;MAPE\;\left(\%\right)=\frac{1}{n}\sum_{i=1}^{n}\frac{\left|y_{i}-{\hat{y}}_{i}\right|}{y_{i}}\;$$
where |∙| represents the absolute operator, and n is the total number for the facial video. The lower value of MAE, RMSE and MAPE means that the calculated FPR estimates and actual PR are closer. As another evaluation method, the similarity between FPR and the actual PR was evaluated through correlation coefficient analysis.

#### 4.2.1. Performance Evaluation for Stable and Unstable Face Detection

Figure 21 shows the results of measuring FPR by applying the Cg signal before and after stable face detection to the method proposed in this paper, and comparing FPR with the actual pulse rate calculated in the PPG device.

We compared FPR with the actual pulse rate calculated in the PPG device. Our experimental results on stable and unstable datasets are presented in Table 1. Except for the color change of the face, noise, such as head tremors, has a tremendous impact on pulse wave and pulse rate calculations, resulting in a lower accuracy of pulse rate measurements.

In the indoor environment dataset, the pulse rate was estimated by the proposed method for the calculated Cg signal by applying stable and unstable face detection methods and comparing with the actual pulse rate calculated by the PPG device. It was confirmed that the pulse rate measurement accuracy was improved.

#### 4.2.2. Performance Evaluation for Indoor Dataset

In this paper, we compared pulse rate calculation results of three additional methods (STFT, ACF, ICA–BPF, weighted average) to evaluate the performance of the proposed method. A graph of the results for each method is shown in Figure 22. With the proposed method of RFPR (robust facial pulse rate), r showed the highest correlation of 0.9472 and FPR (ICA–BPF) showed the lowest correlation of 0.4648. This is because RGB colors may be greatly affected by illumination changes. Our experimental results on the indoor dataset are presented in Table 2.

As shown in Table 2, pulse rates were calculated using Cg signals calculated in both cheeks ($ROI_{1}$, $ROI_{2}$) for our proposed method and three additional methods. Effects of illumination changes were considered by multiplying pulse rates calculated in each method by the weights of left cheek 0.5 and right cheek 0.5. The FPR with the weighted average showed a higher accuracy than the FPR without the weighted average. In addition, the MAPE of our proposed method (RFPR) was improved by 0.51%, from 3.30% to 2.79%.

#### 4.2.3. Performance Evaluation for External Dataset

When performing pulse rate measurement, it is difficult to calculate the pulse rate with a high accuracy using facial videos in an external environment, because noises, such as illumination changes, head tremors, and so on, are more diverse than those in an indoor environment. We compared the calculated pulse rate through facial videos taken from flying drones, driving cars, and outdoors with a PPG device. Our experimental results on the external environment dataset are presented in Table 3.

As shown in Table 3, the proposed method in this paper showed higher accuracy than the other three methods, with a MAPE of 3.34% for outdoors, 6.33% for flying drone, and 6.50% for driving car. In addition, pulse rate measurement results through frequency domain analysis (STFT) were calculated with the lowest accuracy, with a MAPE of 8.02% for outdoors, 8.02% for driving car, and 10.11% for flying drone. It was confirmed that the frequency analysis of pulse-related frequency bands of facial videos are susceptible to noise generated in external environments.

#### 4.2.4. Performance Evaluation for Short-Time Pulse Rate Measurement

As shown in Table 4, the proposed method showed a high accuracy even for short-time pulse rate measurements. MAPE was calculated as 4.08% at 30 s, 4.33% at 15 s, 4.85% at 10 s.

#### 4.2.5. Performance Evaluation on an Indoor Environment according to Distance

We also separately collected facial videos of targets farther than 50 cm using a webcam with an optical zoom function to evaluate pulse rate measurement accuracy according to distance. The distance was 1 m, 5 m, 15 m, 20 m, 25 m, 30 m and 50 m. All 42 facial videos for subjects (three males) were taken with six facial videos according to the distance. Even if a face is detected at 1 m or more, the size of the ROI decreases as the distance increases. For this reason, a normal camera, such as that of a smartphone and a smart device, has limitations in pulse rate measurement. To solve this problem, the optical zoom function of the webcam was applied to expand it to the same height and width as the facial area detected by a smartphone camera. The pulse rate was estimated using the Cg signal calculated from the skin ROI of the enlarged facial video. The MAPE for each distance was calculated to evaluate the performance. The results are shown in Table 5.

## 5. Discussion

In summary, we implemented the stable face detection method and selected the optimal ROI, which enabled the effective analysis of skin color changes. In addition, the robust pulse wave and rate was calculated by using FFT–iFFT with zero-padding. When measuring pulse rates using facial videos collected in indoor and external environments by removing noise from signal distortions, our method is more accurate than other methods (STFT, ACF, ICA–BPF).

We observed possible limitations of our method during experiments. Facial angle changes and facial videos of long distances reduce pulse rate measurement accuracy, because our method calculates the pulse wave/rate from the ROI for arterial blood in the cheek. However, developing a technology that can measure pulse rates using facial videos even in poor conditions would be highly valued when used in diverse fields. Furthermore, our approach is likely to be improved in the future if possible limitations can be considered by combining methodologies associated with deep learning-based face and ROI detection. Since the focus of this research methodology is lightweight, based on cameras of smart devices available in diverse environments, it did not include a deep learning-based methodology. In addition, the smart device environment has a lighter weight than the PC environment, which can result in a lower face detection rate when using deep learning methodology-based face detection and continuous calculation of skin color data. However, we will conduct extensive experiments in the future, including our approach and deep learning methodology.

## 6. Conclusions

In this paper, we proposed a robust pulse rate measurement method in diverse environments. In addition to measuring the pulse rate in an indoor environment, as in existing studies, facial videos were collected from outdoors, cars, and drones. Since the pulse rate calculation using a facial video is affected by noise, such as illumination changes and user movement, accuracy might be lowered. Therefore, we have achieved the precise detection of skin color changes using the stable face detection method and have calculated the pulse rate with a high accuracy by applying the FFT–iFFT method, which could prevent signal distortion and delay. We also applied a stable face detection method. Experimental results showed s higher accuracy after application than before applying the weighted average method considering the amounts of illumination changes applied to both cheeks. The proposed technology can help us not only manage health in an internal environment but also analyze conditions of drivers or victims in real time using cars and drones. Our future work will use cameras higher than 30 fps along with more dataset collection to observe more detailed color changes in facial skin areas. It will be used to measure various bio-signals, such as respiratory, blood pressure, and body temperature, as well as a more accurate pulse rates.

## Figures and Tables

**Figure 1 sensors-22-09373-f001:**
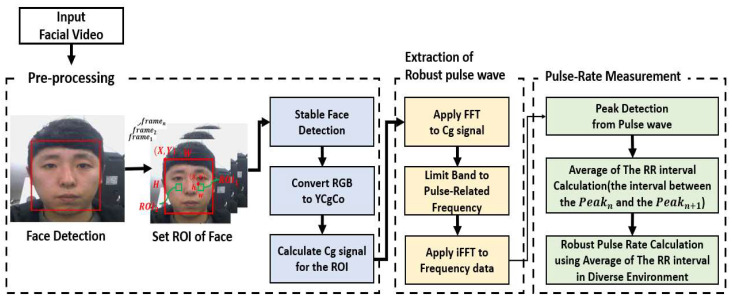
Flowchart of our proposed method.

**Figure 2 sensors-22-09373-f002:**
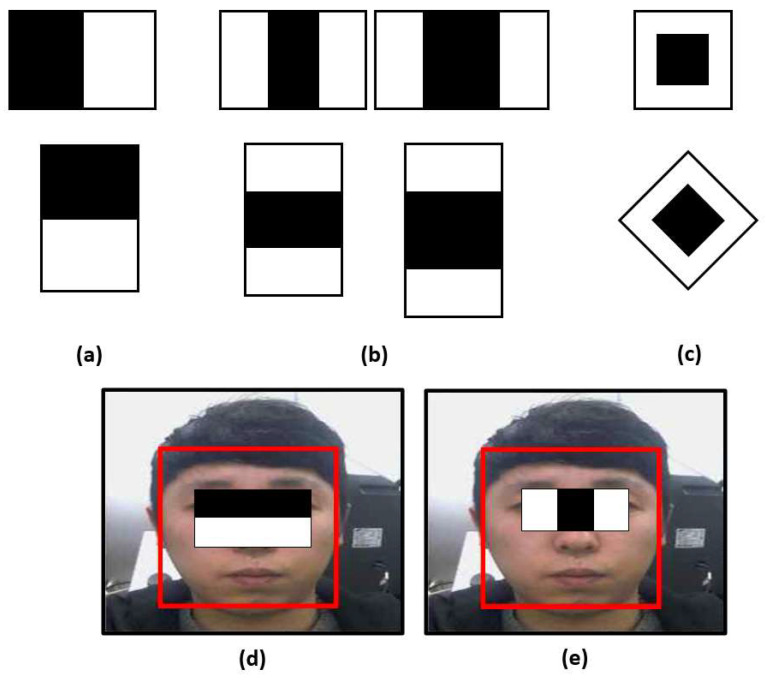
Expended set of Haar-like features: (**a**) edge features, (**b**) line features, (**c**) center-surround features. Face detection based on (**d**) the edge feature of Haar, (**e**) the line feature of Haar.

**Figure 3 sensors-22-09373-f003:**
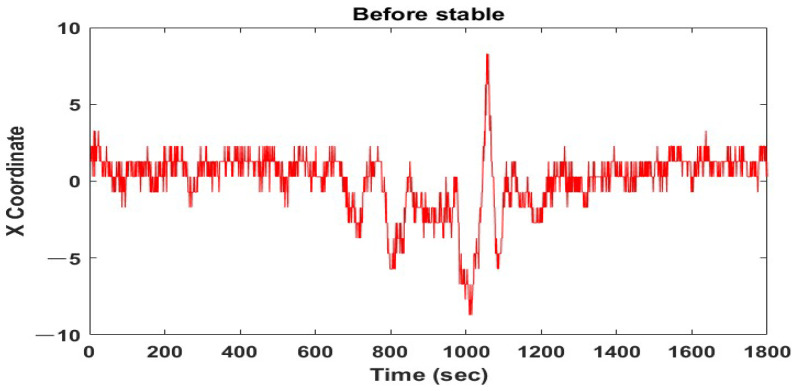
X coordinate position with time.

**Figure 4 sensors-22-09373-f004:**
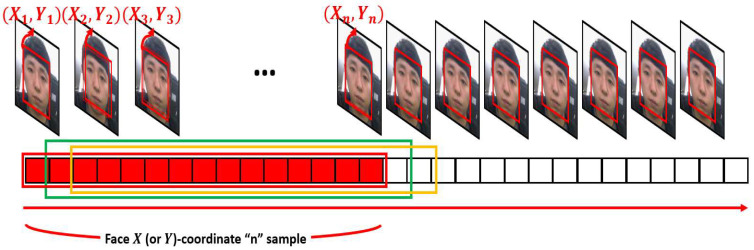
Example of n samples for facial X (or Y)-coordinates.

**Figure 5 sensors-22-09373-f005:**
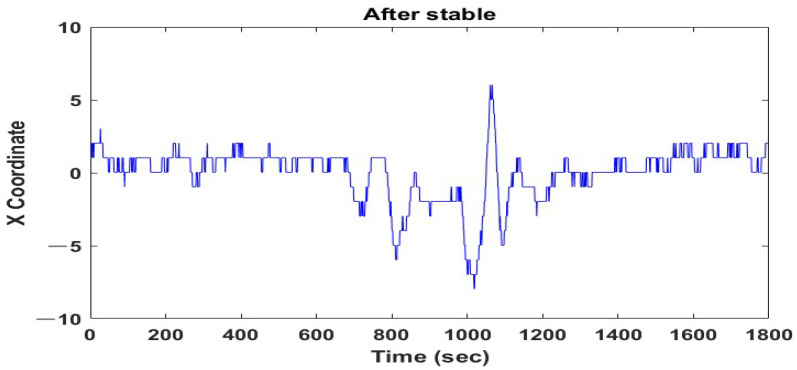
Calibrated coordinate result (*n* = 15).

**Figure 6 sensors-22-09373-f006:**
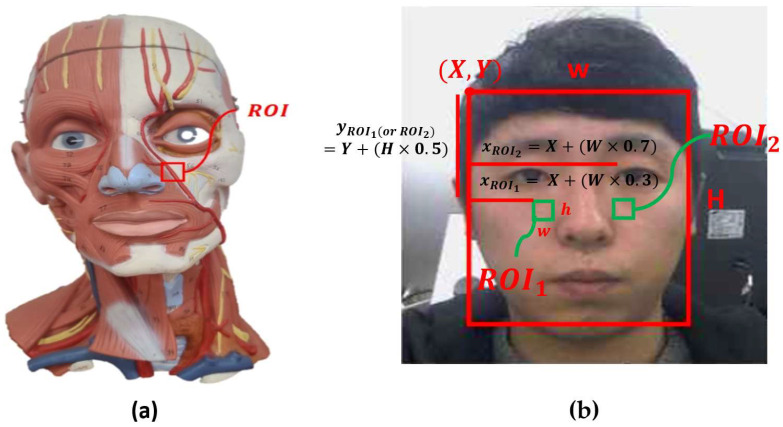
(**a**) Blood vessel and region of interest (red rectangle) of the facial artery. (**b**) Detected cheek ROI (green rectangle) from the facial video.

**Figure 7 sensors-22-09373-f007:**
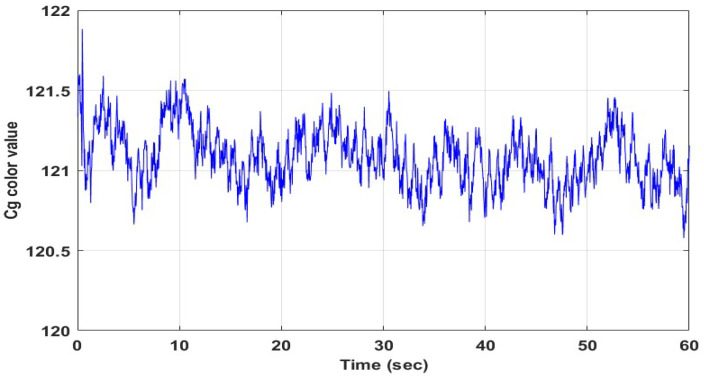
Cg signal recorded for 60 s.

**Figure 8 sensors-22-09373-f008:**
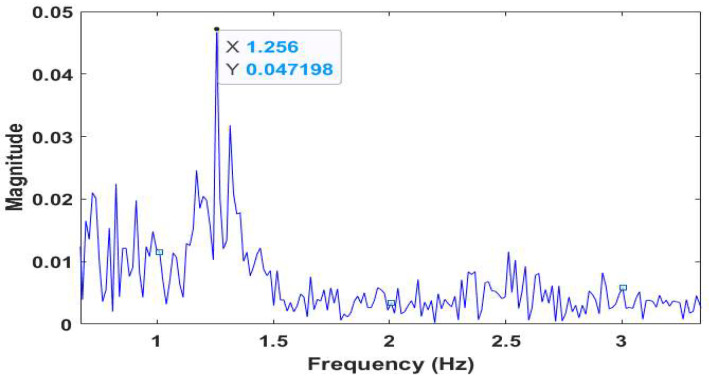
Frequency domain of the Cg signal (Figure 7).

**Figure 9 sensors-22-09373-f009:**
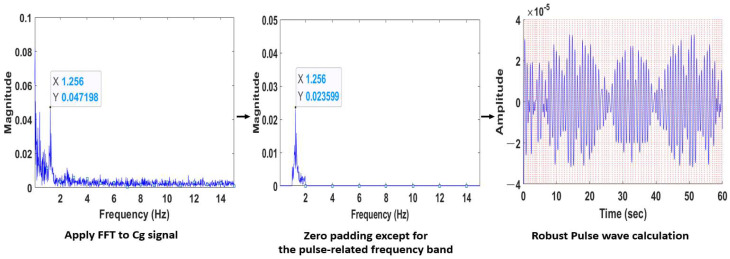
Process of the pulse wave extraction.

**Figure 10 sensors-22-09373-f010:**
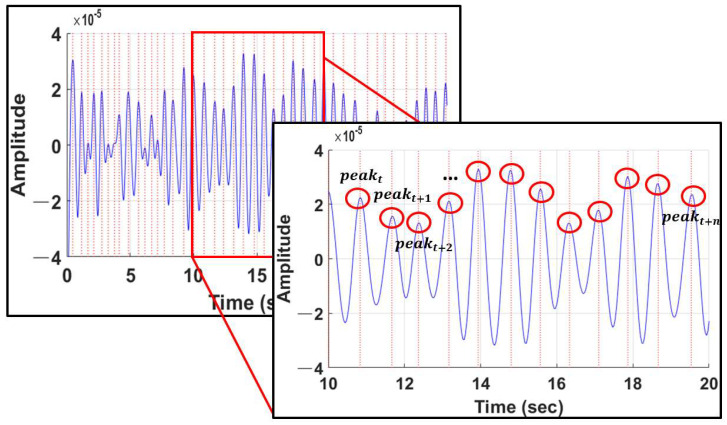
Peak detection of pulse waves.

**Figure 11 sensors-22-09373-f011:**
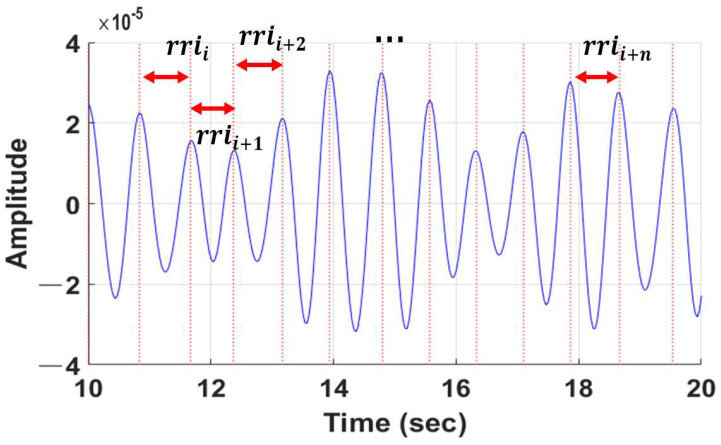
RR interval calculation of the pulse wave.

**Figure 12 sensors-22-09373-f012:**
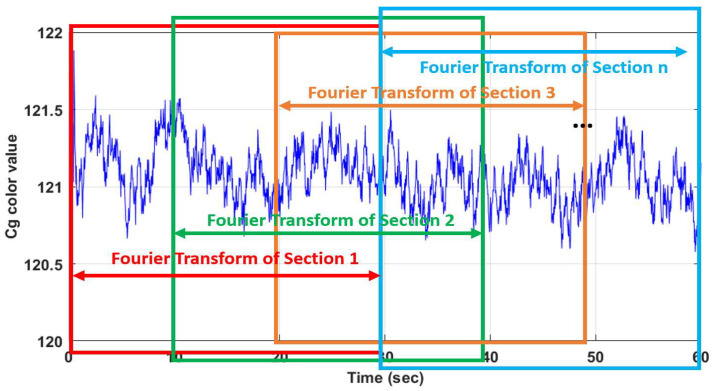
Separation of the Cg signal sections.

**Figure 13 sensors-22-09373-f013:**
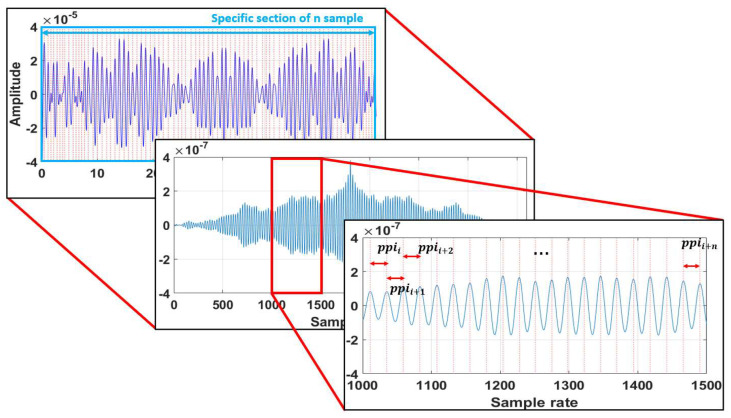
Peak-to-peak interval calculation of the signal with applied ACF to the pulse wave.

**Figure 14 sensors-22-09373-f014:**
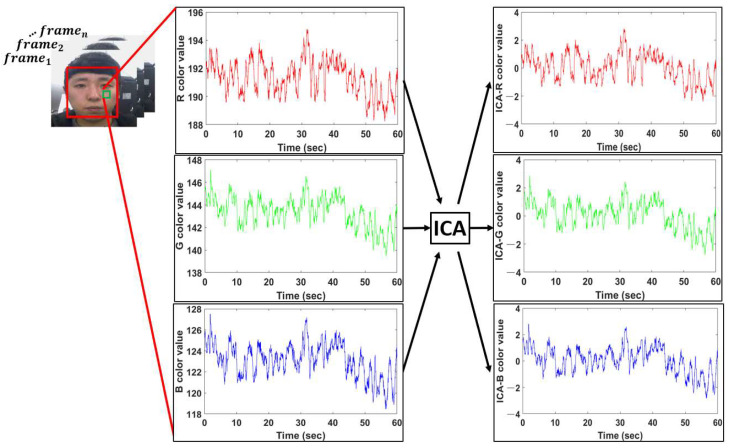
RGB components calculated by applying ICA to RGB color data.

**Figure 15 sensors-22-09373-f015:**
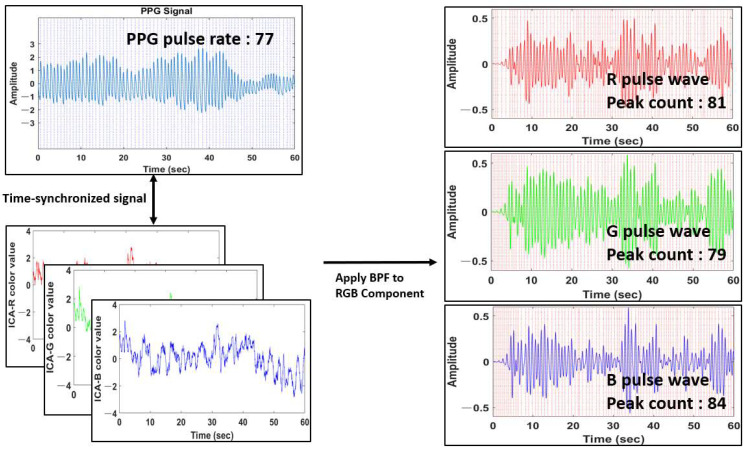
R, G, B pulse waves calculated by applying BPF to the RGB component.

**Figure 16 sensors-22-09373-f016:**
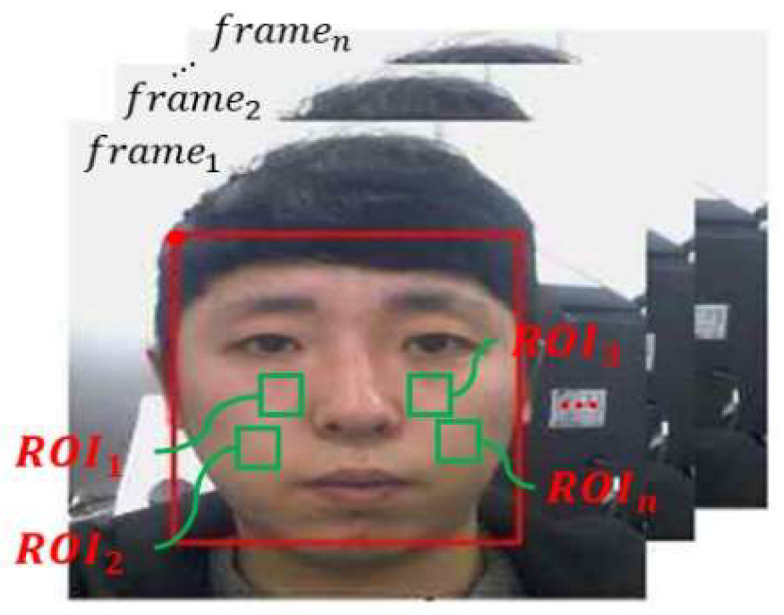
Selection of multiple ROIs.

**Figure 17 sensors-22-09373-f017:**
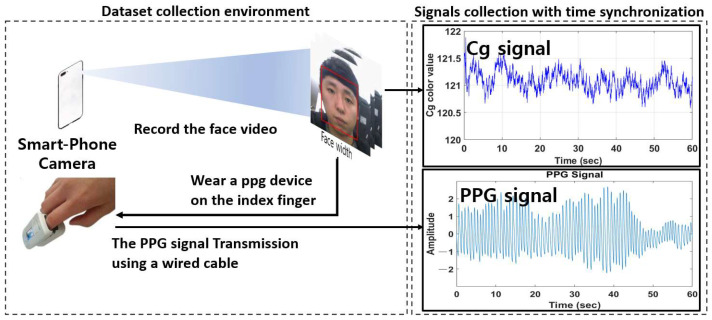
Dataset collection environment for an indoor lab.

**Figure 18 sensors-22-09373-f018:**
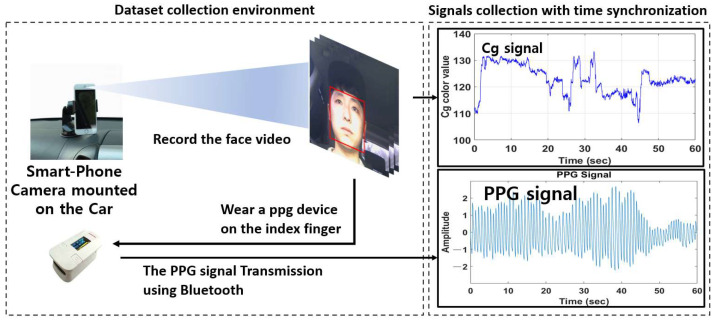
Dataset collection environment when driving a car.

**Figure 19 sensors-22-09373-f019:**
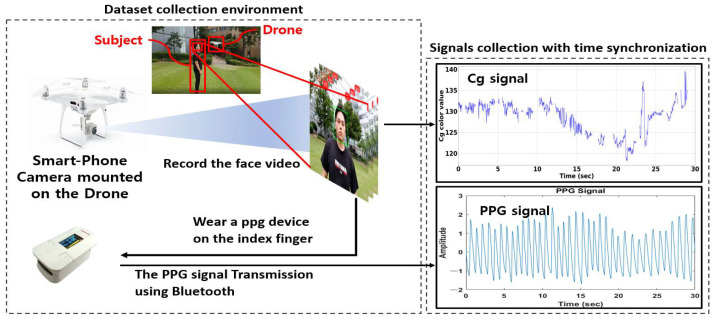
Dataset collection environment when flying a drone (DJI WM3315).

**Figure 20 sensors-22-09373-f020:**
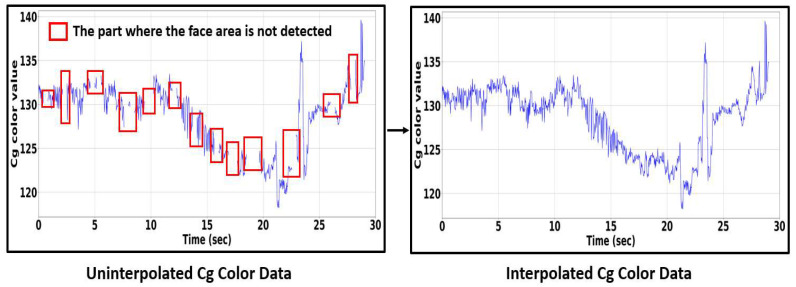
Process of interpolation for a Cg signal.

**Figure 21 sensors-22-09373-f021:**
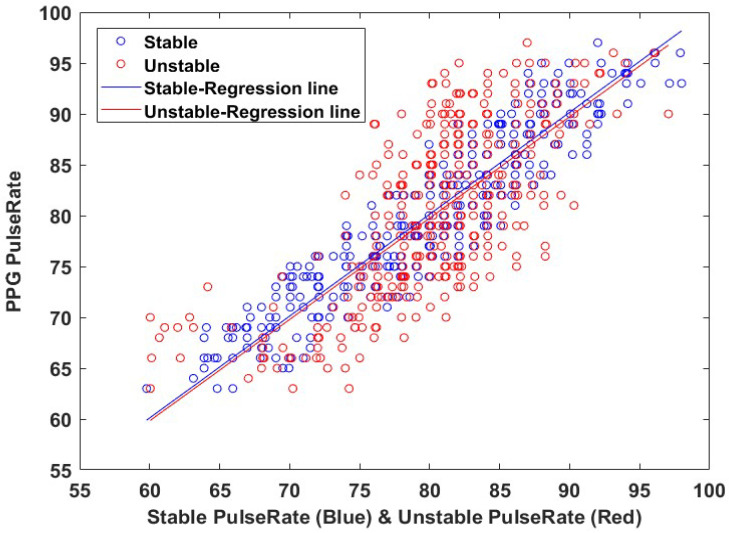
Comparison of FPR calculation results by stable and unstable methods.

**Figure 22 sensors-22-09373-f022:**
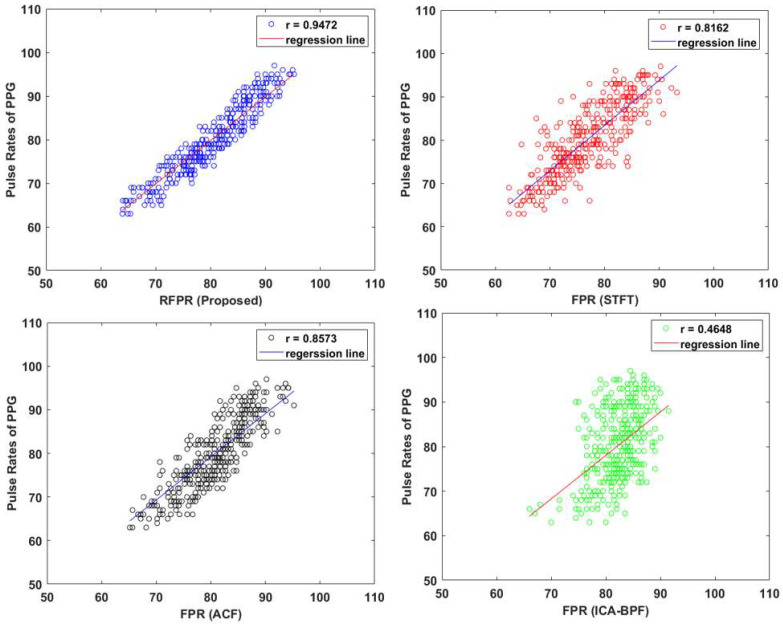
Graphs comparing RFPR with other FPR methods (STFT, ACF, ICA–BPF).

**Table 1 sensors-22-09373-t001:** Comparative results of the FPR on stable and unstable datasets.

Method	Analysis of Different Conditions (Using Cg Data of the Left Cheek)			
	RMSE	MAE	$MAPE\;\left(\%\right)\;$	r
Stable FPR	3.11	2.63	3.30	0.9266
Unstable FPR	5.83	4.93	6.21	0.7016

**Table 2 sensors-22-09373-t002:** Comparative results of the FPR on indoor dataset.

Method		Analysis of Different Conditions			
		RMSE	MAE	$MAPE\;\left(\%\right)$	r
RFRR (Proposed)		3.11	2.63	3.30	0.9266
STFT-FPR		6.14	4.56	5.56	0.7662
ACF-FPR		4.86	3.85	4.87	0.8144
ICA and BPF-FPR		7.76	6.45	8.27	0.4082
Weighted Average	FRPR (Proposed)	2.67	2.22	2.79	0.9472
	STFT	5.73	4.46	5.39	0.8162
	ACF	4.39	3.62	4.61	0.8573
	ICA & BPF	7.50	6.29	8.10	0.4648

**Table 3 sensors-22-09373-t003:** Comparative results of the FPR on external dataset.

External Environment (Weighted Average Application)					
Conditions	Metrics	RFPR	STFT	ACF	ICA&BPF
Outdoor Bench	RMSE	3.36	7.52	3.97	5.98
	MAE	2.89	6.31	3.58	5.12
	MAPE	3.34	8.02	3.85	6.27
	r	0.9033	0.4377	0.8630	0.7229
Driving Car	RMSE	5.81	7.35	5.99	6.31
	MAE	5.17	6.24	5.31	5.88
	MAPE	6.33	8.02	6.36	7.07
	r	0.7426	0.4767	0.7189	0.6122
Flying Drone	RMSE	6.09	9.39	6.57	8.84
	MAE	5.32	7.89	5.96	6.84
	MAPE	6.50	10.11	7.32	8.89
	r	0.7515	0.3698	0.6012	0.3914

**Table 4 sensors-22-09373-t004:** *MAPE* (%) of the short-time FHR on indoor dataset.

Sec\Method	RFPR	STFT	ACF	ICA&BPF (Peak×60/nsec)
60 ssample	2.79	5.39	4.61	8.10
30 ssample	4.08	6.46	4.62	8.32
15 ssample	4.33	7.12	5.43	8.76
10 ssample	4.85	7.24	6.64	9.25

**Table 5 sensors-22-09373-t005:** *MAPE* (%) of the FHR according to distance.

Method\Distance	50 cm	1 m	5 m	15 m	20 m	25 m	30 m	50 m
RFPR	2.83	3.11	3.85	7.07	6.95	8.04	8.23	10.6

## Data Availability

The raw data used to support the findings of this study are available from the corresponding author upon request. However, videos require permission for subjects.

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
