# Peer review of "Robust Pulse Rate Measurements from Facial Videos in Diverse Environments"

_sensors, 2022, doi:10.3390/s22239373_

Round 1

Reviewer 1 Report

English must be improved

“And, Since the majority of..”

Technology:

-          Why not to use face detection based on existing algorithms like media pipe that was trained on millions of faces?

-          Not clear how you balance the different illumination – equation 2 not clear

-          Also, why “Since illumination changes can affect the pulse rate calculation accuracy” – give reference or more explanation

-          Why you calculate the rate by the pulse, and not by moving average on spectrogram? Where the averaging is part of the fft and window length

add also discussion your direct approach results in relation to using deeplearning approach

Author Response

Dear referee

We thank the reviewers for taking the time to review our work and providing their meaningful insights and careful reviews. We have carefully considered all of the comments and this letter gives detailing our changes.

  • We have extensively edited English languages and styles.

    (It can be confirmed in the manuscript containing "Track Changes".)

  • We have added references to unclear descriptions
  • We have added a clear description of the methodology.

Thank you for your consideration of our work

Reviewer 2 Report

The article is devoted to the significant field of cardiovascular system diagnostics by pulse monitoring. The authors develop a highly promising technology for determining the heart rate by non-contact measurements, including those using the camera of an ordinary smartphone.

There are several issues in this article that, from the reviewer's point of view, require revision and corrections.

Author Response

Dear referee

We thank the reviewers for taking the time to review our work and providing their meaningful insights and careful reviews. We have carefully considered all of the comments and this letter gives detailing our changes.

  • We have extensively edited English languages and styles.

    (It can be confirmed in the manuscript containing "Track Changes".)

  • We have added a clear description of the methodology.
  • We combined Figure 7 with Figure 6 and removed Figure 22.
  • We edited most of Figure so that it can be clearly read.

Thank you for your consideration of our work
